# Prediction of the Potential Distributions of *Prunus salicina* Lindl., *Monilinia fructicola*, and Their Overlap in China Using MaxEnt

**DOI:** 10.3390/jof9020189

**Published:** 2023-01-31

**Authors:** Zhe Zhang, Lin Chen, Xueyan Zhang, Qing Li

**Affiliations:** College of Agronomy, Sichuan Agricultural University, Chengdu 611130, China

**Keywords:** brown rot, pathogenic microorganism, environmental variables, ecological modelling, distribution prediction

## Abstract

*Prunus salicina* Lindl. (*P. salicina*) is an essential cash crop in China, and brown rot (BR) is one of its most important diseases. In this study, we collected geographic location information on *P. salicina* and *Monilinia fructicola* (G. Winter) Honey (*M. fructicola*), one of the BR pathogenic species, and applied the MaxEnt model to simulate its potential suitable distribution in China. There have been discussions about the dominant environmental variables restricting its geographical distribution and their overlap. The results showed that the mean temperature of the coldest quarter, precipitation of the warmest quarter, precipitation in July, and minimum temperatures in January and November were the main climatic variables affecting the potential distribution of *P. salicina*, while the coldest quarter, precipitation of the driest month, precipitation of March, precipitation of October, maximum temperatures of February, October, and November, and minimum temperature of January were related to the location of *M. fructicola*. Southern China had suitable conditions for both *P. salicina* and *M. fructicola*. Notably, the overlap area of *P. salicina* and *M. fructicola* was primarily located southeast of 91°48′ E 27°38′ N to 126°47′ E 41°45′ N. The potential overlap area predicted by our research provided theoretical evidence for the prevention of BR during plum planting.

## 1. Introduction

*Prunus salicina* Lindl. (*P. salicina*) is a species of plum in the Rosaceae family. It is also known as Jiaqingzi, Bulin, plum, and Jade Emperor plum. *P. salicina* is cultivated globally, and it is an important temperate fruit [1,2]. *P. salicina* is a deciduous fruit tree that originated in China and has been cultivated for over 3000 years [3]. At present, more than 20 provinces and municipalities in China grow *Prunus*. In 2020, China had 1.95 million hectares of harvested area and produced 6.48 million tons of plums, equivalent to 73.86% of the world’s plum planting area and 52.98% of the world’s plum output, according to the Statistics Department of the World Food and Agriculture Organization [4]. Among the plethora of pathogenic agents attacking *Prunus* crops, brown rot (BR) is the most important disease [5]. It can damage stone fruit trees, such as peaches (*Prunus persica* (L.) Batsch), nectarines (*Prunus persica* var. *nucipersica* (Suckow) C. K. Schneid), plums, apricots (*Prunus armeniaca* L.), and sour cherries (*Prunus cerasus* L.), along with some fruit trees, including apples (*Malus pumila* Mill.), and pears (*Pyrus communis* L.) [6,7,8,9]. BR is caused by *Monilinia* spp., which belongs to the phylum Ascomycota, family Sclerotiniaceae, and order Helotiales [10,11,12,13,14].

BR is widely distributed worldwide, with the most obvious damage in Australia, Asia, America, and Europe [10,15,16,17,18]. The disease can not only harm the buds, branches, flowers, and fruits during the growth stage but also induce fruit canker in the storage period, resulting in the decline of fruit yield and quality and increased economic losses [10,11,19]. The control of BR in the production, storage, and transportation of fruits can not only reduce financial loss but also limit the spread of the *Monilinia* spp. to stop the effect from influencing the full fruit export trade [10,11].

BR was reported in China as early as the 1920s and is generally common in China, with six *Monilinia* spp., including *Monilinia fructicola* (G. Winter) Honey (*M. fructicola*), *Monilinia laxa* (Aderhold & Ruhland) Honey (*M. laxa*), *Monilinia fructigena* (Pers.) Honey (*M. fructigena*), *Monilinia mumeicola* (Y. Harada, Y. Sasaki & Sano) Sand.-Den. & Crous (*M. mumeicola*), *Monilinia polystroma* (G.C.M. Leeuwen) Kohn (*M. polystroma*), and *Monilinia yunnanensis* (M.J. Hu & C.X. Luo) Sand.-Den. & Crous (*M. yunnanensis*), distributed in 22 provincial-level administrative regions [9,11,20,21,22,23]. *M. yunnanensis* was the most widely distributed species in 12 provincial-level administrative regions, mainly harming stone fruit and fruit plants [7]. *M. fructicola* was followed by *M. yunnanensis*, which mainly affects stone fruit plants in Beijing, Shandong, and Hebei Provinces [7]. It was followed by *M. polystroma*, which was mainly distributed in Hebei, Heilongjiang, and Shandong Provinces, and mainly infected fruit plants. *M. mumeicola* has been detected only in peaches, apricot, and Chongqing sour cherry in Hubei Province [5,24,25]. In this study, we pre-investigated 12 plum plantations in Mianyang city, Nanchong city, Luzhou city, Zigong city, Dazhou city, and Yibin city of Sichuan Province from May to June 2022 and detected *M. fructicola* at all sites. However, there were no report of the detection of *M. fructicola* from plum BR in Sichuan province [7,23,25]. We chose this specie as the pathogen marker to explore the geographical distribution of BR in plums by MaxEnt model.

MaxEnt is a widely used software for predicting species geographical distributions, especially when the number of distribution points is uncertain and the correlation between climate and environmental factors is unclear. The MaxEnt model can obtain a high prediction accuracy based only on distribution data [26,27,28], making it possible to analyze and predict the suitable habitat of the pathogen and its host plants. Recently, the MaxEnt model has been widely adopted in predicting plant potential planting areas, animal and plant habitats, invasive plant distribution areas, and quarantine pest prediction [29,30,31,32]. There are a few applications in plant protection, mainly used to analyze the climatic suitability of pests and diseases, predict the invasion possibility of quarantined pests and diseases, and simulate the impact of climate change on the distribution areas of pests and diseases [27,33]. For plant pathology, Wang et al. [34] combined MaxEnt and GIS tools to predict the potentially suitable areas of *Diaphorina citri* under climate change scenarios in China. Galdino’s team first mapped the global scale of the potential risk of the mango’s sudden decline by MaxEnt [35]. In addition to the current climatic scenarios, Ruheili et al. [36] utilized future projected climatic scenarios to eliminate the hotspots and proportions of the areas of witches’ broom disease in Oman. In our pre-investigation, we detected *M. fructicola* from plum fruits in Sichuan province, which is not included in the locations of the previous studies of plum BR. We hypothesized there were some locations with the potential to overlap *P. salicina* and *M. fructicola* in China, but they had not been acknowledged due to the lack of a mature model in the existing literature. Therefore, we collected geographic location information of *P. salicina* and *M. fructicola* by searching databases and the literature, downloaded climate variables from the WorldClim website, and used MaxEnt to simulate the potential suitable distribution of each in China. We evaluated the dominant environmental variables restricting the geographical distribution of *P. salicina* and *M. fructicola*, and assessed their overlap to provide evidence for future research and protection against BR.

## 2. Materials and Methods

### 2.1. Sources of Software and Map

MaxEnt software (version 3.4.1) was downloaded from the Museum of Natural History (https://biodiversityinformatics.amnh.org/open_source/maxent/, accessed on 2 September 2022); Java software was downloaded from the official website (https://www.oracle.com/java/, accessed on 31 August 2022); R (version 4.1.2) and Rstudio software were downloaded from the official websites (https://www.r-project.org/, https://www.rstudio.com/, accessed on 31 August 2022); ArcGIS software (version 10.8.1) came from the website (https://support.esri.com/en/Products/Desktop/arcgis-desktop/arcmap, accessed on 2 September 2022); the base map was provided by the National Meteorological Information Center.

### 2.2. Occurrence Records of P. salicina and M. fructicola

The species distribution data for *P. salicina* and *M. fructicola* were downloaded from the Global Biodiversity Information Facility (GBIF, https://www.gbif.org/, accessed on 3 September 2022) and Centre Agriculture Bioscience International (CABI, https://www.plantwise.org/knowledgebank/, accessed on 3 September 2022), allowing the collection of *P. salicina* and *M. fructicola* occurrence data. The latitude and longitude recorded in the literature for *P. salicina* and *M. fructicola* were determined using Google Earth. Through the above procedure, 457 and 54 distribution data points were obtained for *P. salicina* and *M. fructicola*, respectively (Figure 1).

### 2.3. Climatic Variables Related to P. salicina and M. fructicola

The historical climate data were downloaded from the WorldClim website (https://www.worldclim.org/, accessed on 3 September 2022) with a spatial resolution of 5′. According to the literature review and the pre-investigation of our study, the climate data included 19 bioclimate factors, reflecting the characteristics and seasonal variation of temperature and precipitation with strong biological significance, monthly average precipitation, monthly average maximum, minimum temperature, etc., and the climate index from 1970 to 2000 (Table 1) [8,13,15,16,24,25,36,37,38,39,40,41]. Then, the climate variables were extracted from the administrative zoning map of China as the base map.

For screening modeling variables, correlation analysis of climate data was implemented using ENMtools software to calculate the Pearson coefficient. Initial climate variables and species distribution data were imported into MaxEnt to calculate the initial contribution rate, and variables with very low contribution rates were removed. Suitable environmental variables were screened based on a Pearson coefficient higher than |0.8| (very significant correlation) and the contribution rate.

### 2.4. Construction and Evaluation of the MaxEnt Model

To construct the MaxEnt model, the species distribution data were transferred into csv files, and the tiff variable layer was converted by the format conversion function of ArcGIS into the ASC layer required by MaxEnt software. The species distribution data and climate variables were imported into the software “Sample” and “Environmental layers”, respectively. Response curves for climate variables were created by checking “Create response curves”, the predictions were drawn by “Make pictures of predictions”, and variable importance was measured through Jackknife analysis. Output format and Output file type are set to default values. In the initial model, the “Random test percentage” of test data was set as 25%. Then, the reconstructed models were set to improve the accuracy. “Random seed” was set as a random proportion, “Regularization Multiplier” was set to 1, and the number of “ Replicates” was entered as 10, indicating the model would run 10 times. Other parameters were set to the default software. According to the UN’s Intergovernmental Panel on Climate Change (IPCC)’s explanation of the probability of species presence along with the results of previous research, the suitability grades were divided into four categories and displayed in different colors on the map: highly suitable area (*p* ≥ 0.66, red), moderately suitable area (0.33 ≤ *p* < 0.66, orange), poorly suitable area (0.05 ≤ *p* < 0.33, yellow), and unsuitable area (*p* < 0.05, white) [42,43].

The receiver operating characteristic (ROC) curve output by MaxEnt is one of the most effective and widely-used methods for evaluating the accuracy of niche models by excluding false positive and false negative distribution results [35,44,45,46,47]. The ROC curve is plotted with a false positive rate (1-specific rate) and true positive rate (1-omission rate) as the horizontal and vertical coordinates according to a series of dichotomies. The area under the curve (AUC) is not affected by the incidence of distribution points and the judgment threshold. The value range of AUC is [0, 1]. The closer the AUC is to 1, the greater the correlation between environmental variables and the distribution model and the higher the accuracy of prediction results. AUC values of 0.5–0.7 indicate poor performance. Values of 0.7–0.9 indicate moderate performance, and a value greater than 0.9 indicates high performance [43,48].

### 2.5. Extraction and Analysis of Overlapping Areas in Suitable Areas

The local analysis function of ArcGIS was utilized to extract the grids of overlapping areas of the total suitable areas of *P. salicina* and *M. fructicola*. The distribution in provinces (regions and cities) was calculated according to the grid attributes.

### 2.6. Evaluation of Simulation Results by the MaxEnt Model

According to the local bureau of agricultural statistics, we sampled, identified, and marked the longitude and latitude of the plum plantations using the unit of a county administrative region. After importing the geographic distribution data into ArcGIS, we calculated the distance between the distribution points in the unit grid and the grid centroid and retained the distribution point closest to the centroid. ArcGIS was used to extract the fitness index corresponding to the distribution points in the field survey. The accuracy of the definition of the distribution points in the high-suitability area was 100%, the accuracy of the definition points in the medium-suitability area was 66%, the accuracy of the definition points in the low-suitability area was 33%, and the accuracy of the definition points in the unsuitable area was 0 (Table 2). The evaluation formula is as follows:(1)$$A=\sum_{i}^{n}\frac{x_{i}\times a_{i}}{N}\times 100\%$$

*A*: accuracy; *i*: suitability level; *N*: number of field investigation points; *X_i_*: number of grade i distribution points; *a_i_*: corresponding accuracy of the grade i suitability area.

**Table 2 jof-09-00189-t002:** Comparison table of accuracy and suitable grade.

Suitable Area	Suitable Grade	Accuracy
Unsuitable	1	0%
Low-suitable	2	33%
Middle-suitable	3	66%
Highly-suitable	4	100%

## 3. Results

### 3.1. The Importance of Climate Variables on the Distribution of P. salicina and M. fructicola

The AUC indexes of the reconstructed model for *P. salicina* and *M. fructicola* were 0.954 and 0.961, respectively, indicating their high precision and credibility (Figure 2). According to the jackknife test, the AUC values of five environmental variables (bio11, bio18, prec07, tmin01, tmin11) were all >0.8, indicating that they were the main factors affecting the potential distribution area of *P. salicina* (Figure 3a). The AUC values of bio18 and prec07 were the highest, indicating that the warmest season precipitation and the mean precipitation in July were the most important variables affecting the geographical distribution of *P. salicina* (Figure 3a). Conversely, seven primary factors affected the distribution of *M. fructicola* with AUC values over 0.8 (Figure 3b). The AUC of both tmax10 and tmax11 exceeded 0.92, indicating that the maximum temperatures in October and November were the most essential variables influencing the geographical distribution of *M. fructicola* (Figure 3b). In addition, the mean temperature of the coldest quarter (bio11), precipitation of the warmest quarter (bio18), precipitation of July (prec07), and minimum temperatures of January (tmin01) and November (tmin11) were the main climatic variables affecting the potential distribution of *P. salicina*, while the coldest quarter (bio11), precipitation of the driest month (bio14), precipitation of March (prec03), precipitation of October (prec10), maximum temperatures of February (tmax02), October (tmax10), and November (tmax11), and minimum temperature of January (tmin01) were the factors most related to the location of *M. fructicola* (Figure 3).

The response curves presented the relationship between *P. salicina* and the above-selected environmental variables. Filtered by the response probability >0.66, the average temperature of the coldest quarter (bio11) was from −3.82 to 10.36 °C, the warmest quarterly precipitation (bio18) was from 404.8 to 2200 mm, the average precipitation in July (prec07) was from 131.936 to 450.368 mm, the average minimum temperature in January (tmin01) was from −9.158 to 7.538 °C, and the average minimum temperature in November (tmin11) was from −1.251 to 10.78 °C. These ranges indicated the suitable conditions for *P. salicina* occurrence (Figure 4).

Conversely, the response curves of *M. fructicola* are shown in Figure 5. The most suitable variables and ranges for the distribution of *M. fructicola* were the average temperature of the coldest quarter (bio11) from −3.937 to 11.557 °C, the precipitation of the driest month (bio14) from 24.64 to 308.56 mm, the average precipitation amounts in March (prec03) from 36.0192 to 152.1408 mm and in October (prec10) from 60.2096 to 163.2128 mm, the maximum temperatures in February (tmax02) from 10.027 to 23.887 °C, in October (tmax10) from 13.756 to 22.875 °C, in November (tmax11) from 8.49 to 22.821 °C, and the average minimum temperature in January (tmin01) from −8.928 to 13.714 °C (Figure 5).

### 3.2. Potential Distributions of P. salicina and M. fructicola

The geographical distributions of *P. salicina* and *M. fructicola* in China under the current climate conditions predicted by MaxEnt are shown in Figure 6. The highly suitable areas for *P. salicina* were mainly located in southern China, including Chongqing, Guizhou, Jiangsu, most of Zhejiang, most of Anhui, most of Guangxi, Yunnan, Fujian, southeastern Sichuan, northwestern Hunan, southern Henan, and Shandong, as well as in Tibet in northern Guangdong and Jiangxi, and in central Taiwan (Figure 6a). The highly suitable area was 148.54 × 104 km^2^, while the total suitable area was 554.14 × 104 km^2^, accounting for 15.41% and 57.50% of China’s land, respectively.

However, there was a smaller, highly suitable area for *M. fructicola*, which was mainly located in the Yunguichuan Plateau and Chongqing (Figure 6b). The other areas of southern China were marked as moderately suitable for *M. fructicola*, including Henan, Jiangxi, Jiangsu, Zhejiang, Hunan, Guangxi, and Anhui (Figure 6b). There was a 58.96 × 104 km^2^ area marked as highly suitable for *M. fructicola*. Its total suitable area covered 382.63 × 104 km^2^, accounting for 6.12% and 39.71% of China’s land area (Figure 6b), respectively.

### 3.3. The Overlap Area between P. salicina and M. fructicola

The overlap area between *P. salicina* and *M. fructicola* is mostly located southeast of line 91°48′ E 27°38′ N to 126°47′ E 41°45′ N (Figure 7a). Except for Xinjiang, Tibet, Gansu, Inner Mongolia, Heilongjiang, and Jilin, it covers almost all of the land in southern China, with a total area of 380.125 × 104 km^2^ (Figure 7a). This accounted for 68.60% and 99.35% of the suitable area for *P. salicina* and *M. fructicola* (Figure 7b,c), respectively, showing a high degree of coincidence. The distribution area of only *P. salicina* was marked in Heilongjiang, Jilin, a small part of Inner Mongolia, Qinghai, and Tibet (Figure 7a). Conversely, the distribution area of only *M. fructicola* was scattered in Tibet and Xinjiang, with an area of 2.502 × 104 km^2^, accounting for 0.65% of its suitable area (Figure 7a).

### 3.4. Independent Sample Evaluation

Field investigation of species distribution is the most direct and reliable way to verify the model. In this study, the accuracy of simulation results was further verified by sample collection from several plum plantations in Sichuan Province. We filtered 12 plum plantations (Table 3), among which 10 were located in highly suitable areas, 2 in moderately suitable areas, and 0 in poorly suitable and unsuitable areas. The representative figure of the plum fruit with BR and cultured *M. fructicola* is shown in Figure 8. According to Formula (1), the calculation accuracy was 94.33%, indicating that the model’s simulation performance was good.

## 4. Conclusions and Discussion

Based on the MaxEnt model, we concluded the key environmental variables affecting the distribution of *P. salicina* and *M. fructicola* in China. The average temperature of the coldest quarter (−3.82~10.36 °C), the warmest quarterly precipitation (404.8~2200 mm), the average precipitation in July (131.936~450.368 mm), the average minimum temperature in January (−9.158~7.538 °C), and the average minimum temperature in November (−1.251~10.78 °C) were the suitable conditions for *P. salicina* occurrence. The essential variables of *M. fructicola* were the average temperature of the coldest quarter (−3.937~11.557 °C), the precipitation of the driest month (24.64~308.56 mm), the average precipitation amounts in March (36.0192~152.1408 mm) and in October (60.2096~163.2128 mm), the maximum temperatures in February (tmax02) from 10.027 to 23.887 °C, in October (13.756~22.875 °C), and in November (8.49~22.821 °C), and the average minimum temperature in January (−8.928~13.714 °C) (Figure 4 and Figure 5). The key environmental variables predicted that *P. salicina* was highly suitable to southern China, including Chongqing, Guizhou, Jiangsu, most of Zhejiang, most of Anhui, most of Guangxi, Yunnan, and Fujian, southeastern Sichuan, northwestern Hunan, southern Henan, Shandong, and in Tibet in northern Guangdong and Jiangxi, along with central Taiwan, and *M. fructicola* was mainly located in the Yunguichuan Plateau and Chongqing in China (Figure 6). Nevertheless, the overlap area of *P. salicina* and *M. fructicola*, which was at risk of plum BR infected by *M. fructicola*, was mostly located southeast of line 91°48′ E 27°38′ N to 126°47′ E 41°45′ N in China (Figure 7).

*Monilinia* spp. easily colonize on wounds formed by fruit rupture and produce a large number of conidia from the dead fruit or diseased remnant to infect the flowers or young fruit of the tree when there is enough rain in the spring [37,38]. In our study, the suitable temperature and humidity in winter and spring provided the environmental conditions for *M. fructicola* survival (Figure 5). *M. fructicola* was first found in China in 2003 and has been distributed in Beijing, Shandong, Hebei, and other major stone fruit-producing areas [41]. The distribution areas of *M. fructicola* around China were Gansu, Yunnan, Chongqing, Hubei, Jiangxi, Fujian, Zhejiang, Shanghai, Shandong, Hebei, Beijing, and Liaoning [5,20,21,23,39]. Based on MaxEnt, our prediction of the suitable areas of *M. fructicola* was consistent with the conclusions of previous research (Figure 6b). In this study, the unsuitable areas of *M. fructicola* in China included Northeast China, North China, Northwest China, and the Qinghai-Tibet Plateau, which may stem from the extremely cold and long-duration winters and drought in these areas.

To date, most of the research on BR in China has focused on peach BR. Only a few studies have focused on BR in other stone fruit trees, especially the plum [20,21,24,40,41]. *M. fructicola* was the second wildly distributed species, followed by *M. yunnanensis* in China, and mainly affects stone fruit plants in Beijing, Shandong, and Hebei Provinces [7,39]. However, the detected *M. fructicola* from the plum was only in Beijing, Shandong, Chongqing, and Yunnan [7,23,25]. In recent research, two other species of *Monilinia* spp., *M. fructigena* and *M. polystroma*, have been detected in the plum in China [39,41]. Several studies have suggested that *M. laxa*, *M. fructigena*, and *M. fructicola* have a close genetic relationship, which may contribute to the errors in early molecular sequencing identification [14,23,49,50,51]. However, our research predicted the overlap area of *P. salicina* and *M. fructicola* included, and was larger than the existing records, and only detected *M. fructicola* from the plum fruits, indicating the limitation of sample collection and the deficiency of research on plum BR. According to our prediction, provinces, including Sichuan, Guizhou, Guangxi, Guangdong, Hunan, Hubei, Henan, Anhui, Jiangxi, Jiangsu, Fujian, Zhejiang, and Taiwan, were in the overlap area and had highly suitable area for *P. salicina* planting, but didn’t have the detection of *M. fructicola* from previous studies (Figure 6a and Figure 7). Above provinces could be recognized as the potential disease areas of the plum BR caused by *M. fructicola*, suggesting the need of sampling verification in further researches and the prevention during the plum planting in these areas.

The niche model is based on the assumption that the niche demand of a species is conservative. Factors such as sample size, spatial scale, and climate variables will affect the prediction ability and stability of the model [52]. The species distribution data used in this study were mainly from databases and literature reviews, which ensured the operational requirements of the model, but there were no omissions. In the data obtained by database retrieval and literature review, some distribution points lacked latitudes and longitude and were determined by searching place names through coordinate positioning software, resulting in geographical errors. Furthermore, the occurrence and prevalence of plant pests and diseases are not only affected by climate but also closely related to host conditions, the species and quantity of natural enemies, and the frequency of orchard medication. Environmental factors affecting the distribution of host plants include climate, soil type, vegetation type, topographic factors, variety type, human activities, and socioeconomic structure [53,54,55]. Due to unknown changes in many future factors, and to reduce the complexity of the model, other factors were not included in the environmental variables in this study. It can be speculated that the niche predicted by the MaxEnt model is wider than the actual niche it occupies. In the next step, in addition to the influence of climate factors, the interaction between target species and other factors, the lagged phenomenon of climate change on species distribution, the changes in soil type and vegetation type, and the influence of human activities should also be considered to improve the prediction effectiveness of the model.

## Figures and Tables

**Figure 1 jof-09-00189-f001:**
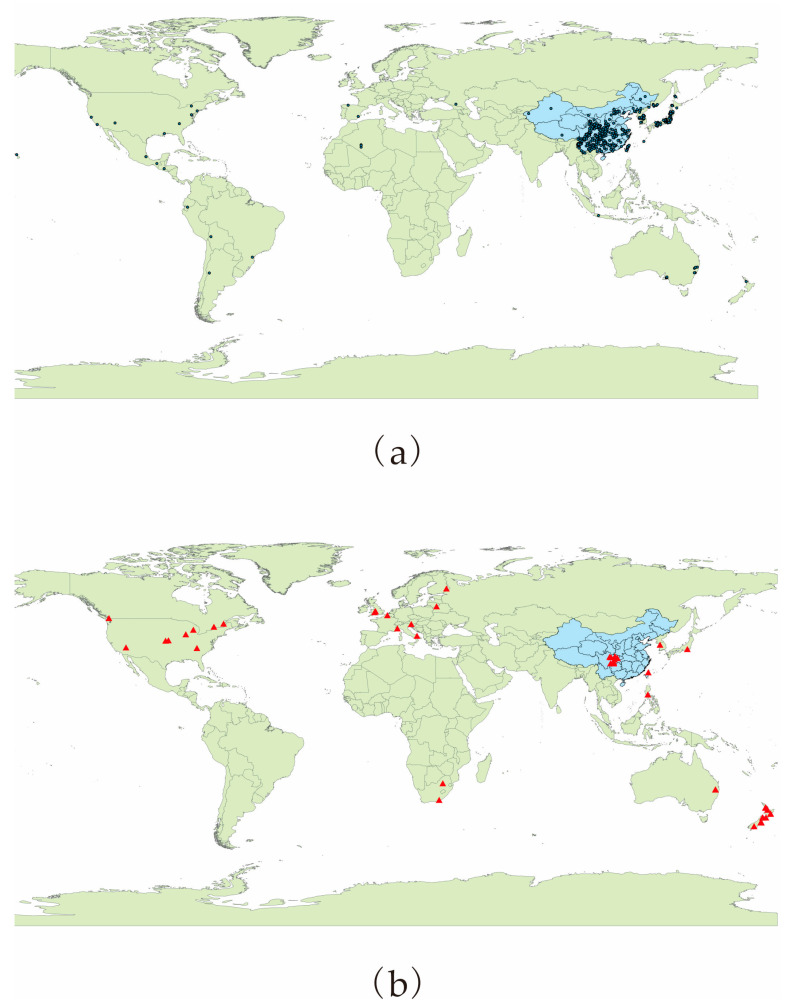
Species occurrence records of *P. salicina* and *M. fructicola*. Notes: The black circles for *P. salicina* (**a**) and red triangles for *M. fructicola* (**b**) indicate distribution points.

**Figure 2 jof-09-00189-f002:**
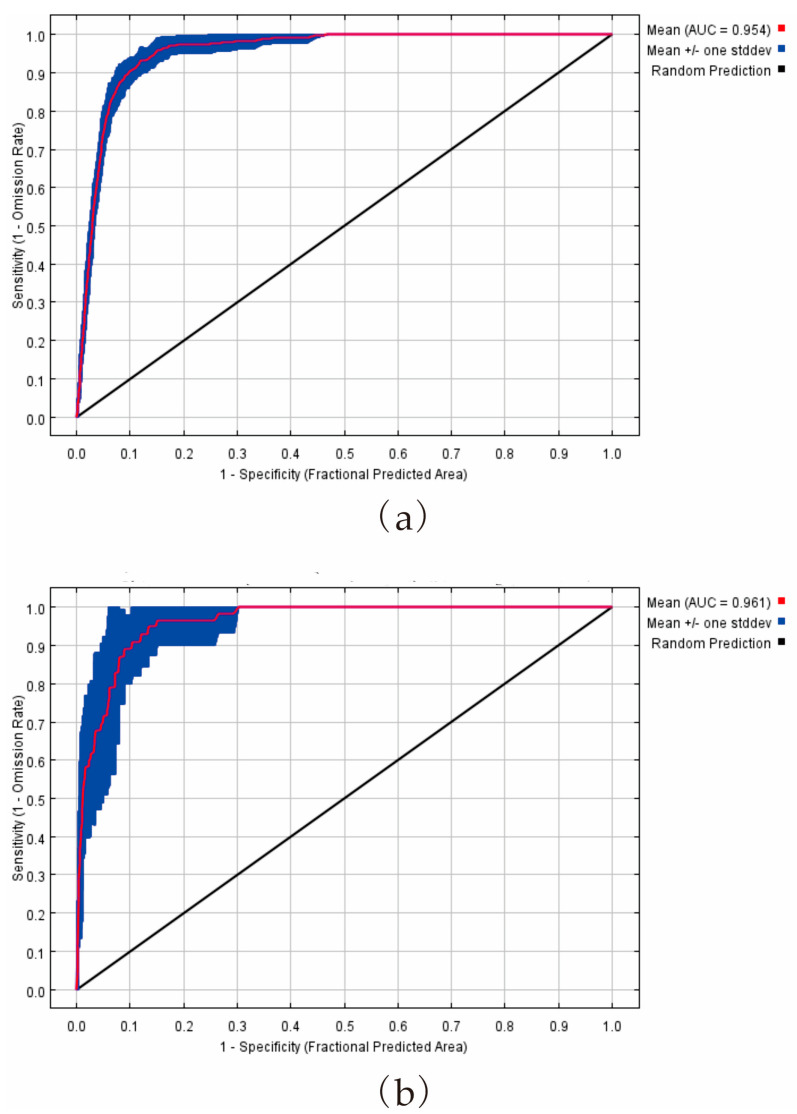
ROC curve and AUC values for the reconstructed model for *P. salicina* (**a**) and *M. fructicola* (**b**).

**Figure 3 jof-09-00189-f003:**
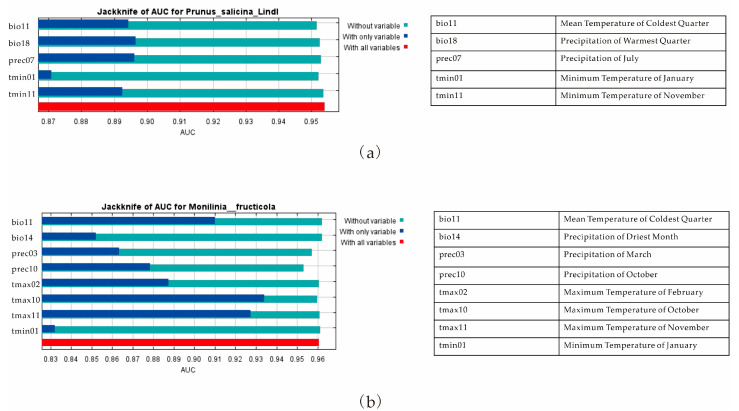
The importance of climate variables on the distribution of *P. salicina* (**a**) and *M. fructicola* (**b**).

**Figure 4 jof-09-00189-f004:**
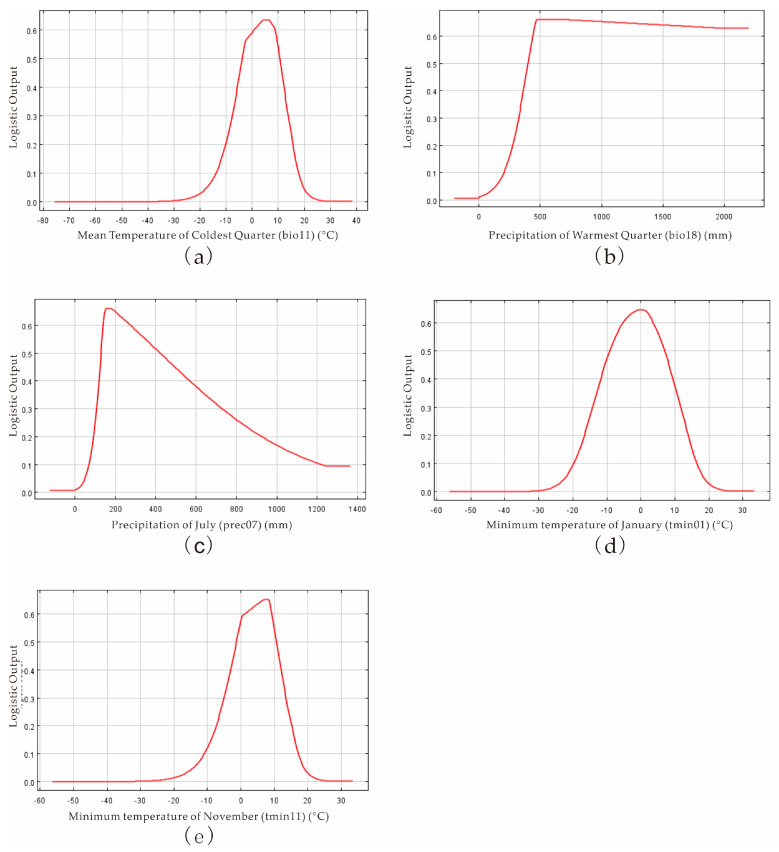
Relationship between distribution area of suitable areas and dominant climate variables ((**a**) mean temperature of coldest quarter; (**b**) precipitation of warmest quarter; (**c**) precipitation of July; (**d**) minimum temperature of January; (**e**) minimum temperature of November) of *P. salicina*.

**Figure 5 jof-09-00189-f005:**
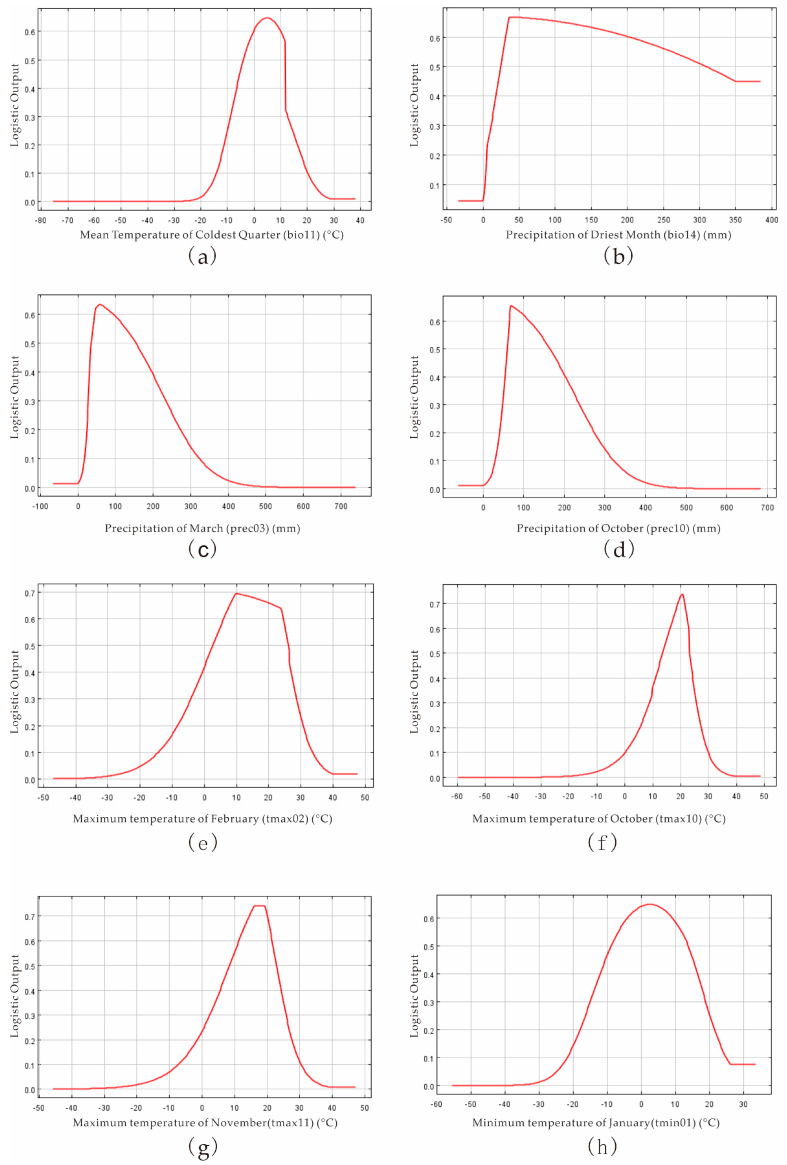
Relationship between distribution area of suitable areas and dominant climate variables ((**a**) mean temperature of coldest quarter; (**b**) precipitation of driest month; (**c**) precipitation of March; (**d**) precipitation of October; (**e**) maximum temperature of February; (**f**) maximum temperature of October; (**g**) maximum temperature of November; (**h**) minimum temperature of January) of *M. fructicola*.

**Figure 6 jof-09-00189-f006:**
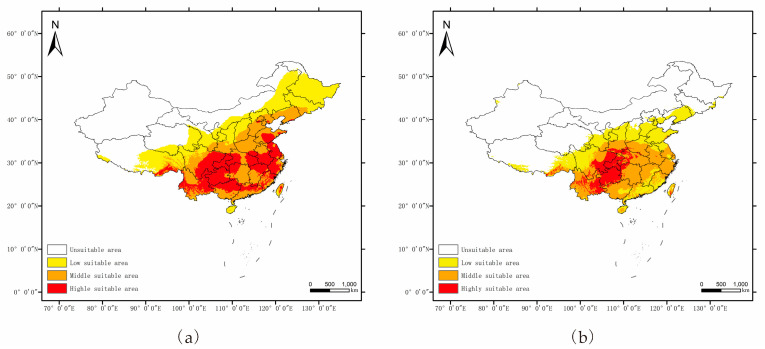
Potential distribution of *P. salicina* (**a**) and *M. fructicola* (**b**) in China.

**Figure 7 jof-09-00189-f007:**
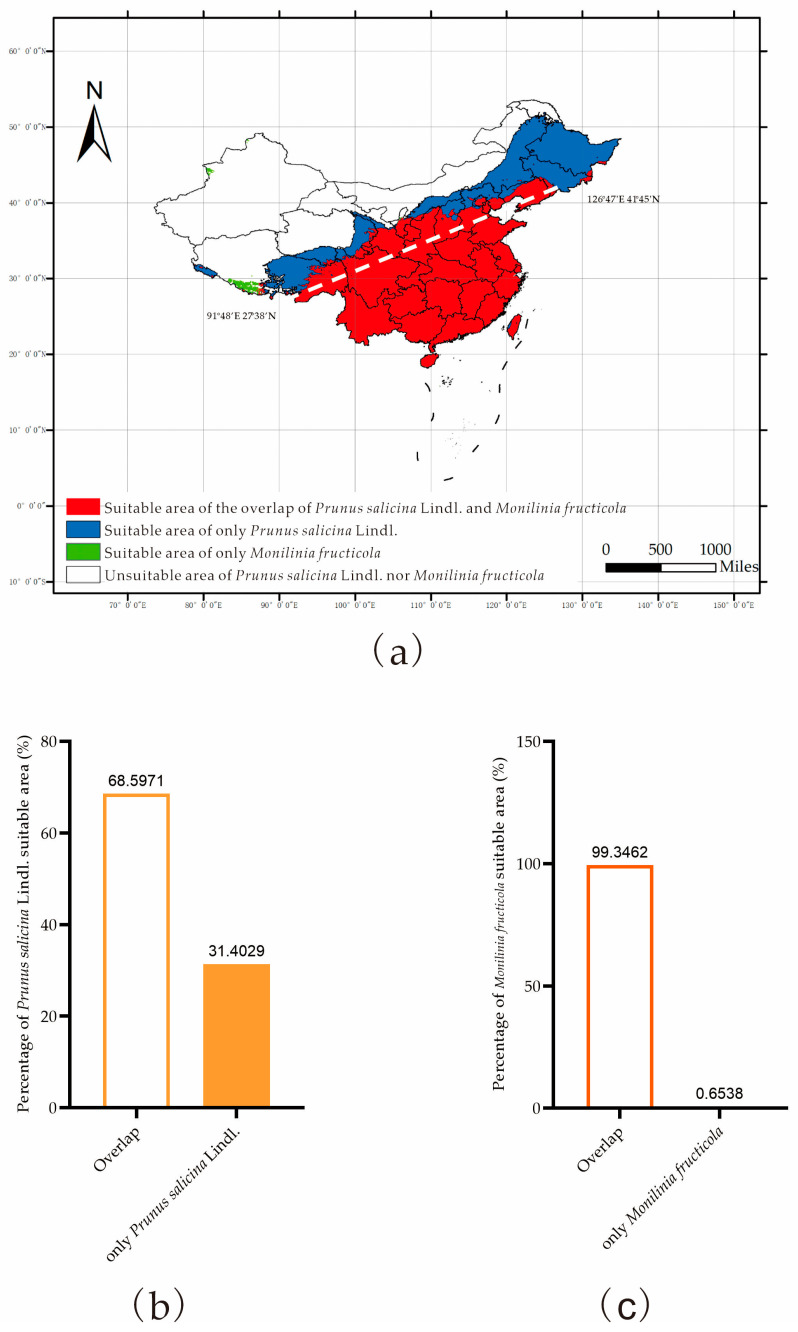
Hot spot map of distribution consistency between *P. salicina* and *M. fructicola* in China (**a**) and their percentage of suitable areas (**b** and **c**).

**Figure 8 jof-09-00189-f008:**
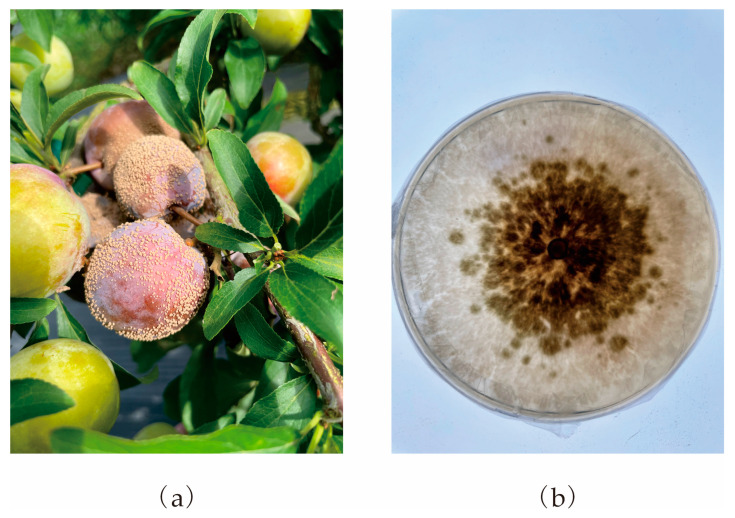
The representative figure of the plum fruit with BR (**a**) and cultured *M. fructicola* (**b**).

**Table 1 jof-09-00189-t001:** List of environmental variables.

Abbreviations	Variables
bio1	Annual Mean Temperature
bio2	Mean Diurnal Range (Mean of monthly (max temp–min temp)
bio3	Isothermality (bio2/bio7) (* 100)
bio4	Temperature Seasonality (standard deviation *100)
bio5	Max Temperature of Warmest Month
bio6	Min Temperature of Coldest Month
bio7	Temperature Annual Range (bio5–bio6)
bio8	Mean Temperature of Wettest Quarter
bio9	Mean Temperature of Driest Quarter
bio10	Mean Temperature of Warmest Quarter
bio11	Mean Temperature of Coldest Quarter
bio12	Annual Precipitation
bio13	Precipitation of Wettest Month
bio14	Precipitation of Driest Month
bio15	Precipitation Seasonality (Coefficient of Variation)
bio16	Precipitation of Wettest Quarter
bio17	Precipitation of Driest Quarter
bio18	Precipitation of Warmest Quarter
bio19	Precipitation of Coldest Quarter
tmin	Minimum Temperature of Each Month
tmax	Maximum Temperature of Each Month
tmean	Mean Temperature of Each Month
prec	Precipitation of Each Month

**Table 3 jof-09-00189-t003:** List of location, longitude, and latitude of plum sample points.

Sample Points	Longitude	Latitude
Jiangyou City, Mianyang City	104.5220	31.5158
Jian Town, Jialing District, Nanchong City	106.0277	30.5139
Heshi Town, Dachuan District, Dazhou City	107.2543	31.1019
Mijiaba Town, Dachuan District, Dazhou City	107.4249	31.1017
Xianshi Town, Hejiang County, Luzhou City	105.7398	28.7154
Yaoba Town, Hejiang County, Luzhou City	105.6556	28.7357
Fawangsi Town, Hejiang County, Luzhou City	105.6664	28.7086
Lijiang Town, Hejiang County, Luzhou City	105.7994	28.7639
Zitong Town, Jiangan County, Yibin City	105.6664	28.7086
Dacheng Town, Pingshan County, Yibin City	104.2765	28.7734
Banqiao Town, Fushun County, Zigong City	104.7000	29.1688
Yuedong Town, Cangxi County, Guangyuan City	106.2668	31.9840

## Data Availability

All data sets generated and/or analyzed in the current study are available from the corresponding author upon reasonable request.

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
