# Peer review of "Prediction of the Potential Distributions of Prunus salicina Lindl., Monilinia fructicola, and Their Overlap in China Using MaxEnt"

_jof, 2023, doi:10.3390/jof9020189_

Round 1

Reviewer 1 Report

Dear colleagues,

This review is concerning a research work entitled “Prediction of the potential distributions of Prunus salicina Lindl, Monilinia fructicola, and their overlap in China using MaxEnt, by Zhe Zhang, Lin Chen, Xueyan Zhang and Qing Li. As detailed and numerous data increasing the knowledge of fungi and plants, I recommend it for an international audience in this journal, however several points have to be precised and a major revision is requested.

Please notice that in order to bring a broad audience to this article and to this journal, for specialists and non-specialists, the six major points of my comments (at the beginning) are very important (mandatory…) for a suitable value of the article. Minor points are also enhanced at the end of this review.

I deeply hope to see this good article published soon,

The six major points are:

1-1     The main point concerns the too synthetized data presented more as conclusions than as experimental procedures. In details, 1/ for figure 4, it is not clear if the variables of the two diagrams were selected before from the whole set of variables or not by a first Maxent (I suppose ?), it seems necessary to insert in this paper (as supplementary material ? see with the editor) the diagrams with all variables of these Maxent(s), then explain your selection for the few (but very important) variables of figure 4 for the two taxa; moreover insert in a caption below the figure the meaning of each variable, to be understood rapidly by readers. 2/ In details too and for the discussion part, about Maxent use, see the tutorial (or equivalent) in  https://www.gbif.org/fr/tool/81279/maxent

then:  https://www.gbif.org/fr/tool/81279/maxent

then: “tutorials”   “tutorials data”   “current tutorial in English”

then:  https://biodiversityinformatics.amnh.org/open_source/maxent/Maxent_tutorial_2021.pdf

in this tutorial See « analysis of variables contributions » with the table of all variables data, see also the different “jackknife” and read particularly the discussion(s) about the three colors (that you use but do not discuss) of the diagrams. Your discussion will be much more supported than it is in the present state.

2-2     The second important point concerns the numerous and interesting results and discussion parts that have to be re-organised. 1/ Restrict the results part to the only description of values and etc.., and remove all sentences which belong actually to the discussion part; check these in all sentences of the results. 2/ In the discussion part, the first paragraph ("the jackknife...") looks  actually like a conclusion, the second paragraph ("Maxent...") looks like part of the  introduction, the third and fourth paragraphs are too general and lack of data (values, comparisons…) for a real (and very fruitful) discussion of this present interesting research.

3-3    As I am involved in taxonomy I am very sensible to correct taxa names, which should be inserted at least the first time they appear in the text with their author(s). So from the introduction, insert latin names in italics (and author(s)) of all plants cited (for instance peaches, plums, apricots…); check them one by one all through the text, even if cited only once. Do the same for all fungi. Use international Plant Names Index (IPNI) https://www.ipni.org/) for plants, or equivalent. For fungi use https://www.mycobank.org/ or equivalent.

4-4     In order to be more attractive especially for non-specialists, as you mention visits of plantations for your research, photo(s) of the plant (various organs, infected or not) and fungi are necessary for this journal devoted to fungi, especially for non-specialists.

5-5     In a conclusion (? or end of discussion), it would be great, as an added value to these interesting data, in order to provide to farmers or others a practical strategy in their fields, to discuss that for instance, as in figure 7 the potential (high suitable) area of the fungus is smaller than the plum tree (?), does it mean that the fungus is susceptible to invade more localities of plum trees in the future? Although it is not directly your paper, is there any possibility of strategy (advisements for the farmers...) against this? Add this point in the abstract as it may be very attractive for the readers.

6-6     References already taken in account by the authors are of real interest, however checking briefly in the word of science WOS and scilit (from mdpi) with the key-words of the abstract, other references (among them very recent ones) appear and references should be once more selected and used (if relevant…) in order to provide a larger view of this interesting research. Among these are the followings:

[1-13]

1.         Arroyo, F.T.; Camacho, M.; Daza, A. First Report of Fruit Rot on Plum Caused by Monilinia fructicola at Alcala del Rio (Seville), Southwestern Spain. Plant Dis 2012, 96.

2.         Latorre, B.A.; Díaz, G.A.; Valencia, A.L.; Naranjo, P.; Ferrada, E.E.; Torres, R.; Zoffoli, J.P. First Report of Monilinia fructicola Causing Brown Rot on Stored Japanese Plum Fruit in Chile. Plant Dis 2014, 98.

3.         Shen, J.C.; Zhang, Z.H.; Liu, R.; Wang, Z.H. Ecological restoration of eroded karst utilizing pioneer moss and vascular plant species with selection based on vegetation diversity and underlying soil chemistry. International journal of phytoremediation 2018, 20.

4.         Acuña, C.V.; Rivas, J.G.; Brambilla, S.M.; Cerrillo, T.; Frusso, E.A.; García, M.N.; Villalba, P.V.; Aguirre, N.C.; Martínez, M.C.; Hopp, E.H.; et al. Characterization of Genetic Diversity in Accessions of Prunus salicina Lindl: Keeping Fruit Flesh Color Ideotype While Adapting to Water Stressed Environments. Agronomy 2019, 9.

5.         Guven, H.; Everhart, S.E.; Angelini, R.M.D.M.; Ozkilinc, H. Genetic diversity assessments of brown rot pathogen Monilinia fructicola based on the six simple sequence repeat loci. J Plant Dis Protect 2021, 128.

6.         Monilinia fructicola (brown rot). Plantwiseplus Knowledge Bank 2022, Species Pa. doi:10.1079/pwkb.species.34746   https://www.scilit.net/article/b1f966077de18952e72f22b2304392ec

7.         Angelini, R.M.D.M.; Landi, L.; Raguseo, C.; Pollastro, S.; Faretra, F.; Romanazzi, G. Tracking of Diversity and Evolution in the Brown Rot Fungi Monilinia fructicola, Monilinia fructigena, and Monilinia laxa. Frontiers in Microbiology 2022, 13.

8.         Cabi. Monilinia fructicola (brown rot). 2022. https://www.scilit.net/article/3825f746ea834383d8f932a4b54ce8ca

doi:10.1079/cabicompendium.34746

9.         Kotiyal, A.; Nautiyal, A.R.; Joshi, V.C. Propagation of Prunus salicina (L.) Santa Rosa: Effect of planting times and auxin levels on success of stem cuttings under mist environment. Ecology, Environment and Conservation 2022.

10.       Li, S.C.; Wang, Y.B.; Wu, F.; Xiao, L.H.; Peng, W.W.; Xiang, M.L.; Chen, J.Y.; Chen, M. First Report of Pyrus pyrifolia Cuiguan Fruit Rot Caused by Monilinia fructicola in Southern China. Plant Dis 2022, 106.

11.       Patel, J.S.; Tian, P.; Navarrete-Tindall, N.; Bartelette, W.S. Occurrence of brown rot of wild plum caused by Monilinia fructicola in Missouri. Plant Hlth Prog 2022.

12.       Silan, E.; Ozkilinc, H. Phylogenetic divergences in brown rot fungal pathogens of Monilinia species from a worldwide collection: inferences based on the nuclear versus mitochondrial genes. BMC Ecology and Evolution 2022, 22.

13.       Tshikhudo, P.P.; Nnzeru, L.R.; Munyai, T.C. Monilinia fructicola intercepted on Prunus spp. imported from Spain into South Africa between 2010 and 2020. South African journal of Science 2022, 118.

 Minor points are:

1 in the introduction, line 36 is not very clear as Monilinia is already a genus (?) and you write that Arrocytidae is a genus, reword this part. Check also again the family name as apparently the articles and internet sites are not convergent about the affinity you write;

2 in the introduction line 47, change “Monilinia app.” for “Monilinia spp.”;

3 figure 1 is too small, even on my large computer screen I can see hardly see the occurences of the two taxa;

4 in table 1, put "variables" instead of "viriables;

5 for figure 2, put a caption explaining briefly the different steps and their relationship, especially for non-specialists;

6 figures 3 and 4 are too small too;

7 in figures 5-6, we cannot read the vertical legends on the left side;

8 in 3.2, you put twice "Fig. 7" in the second line;

9 in figure 8, we cannot see the legends of colours on the bottom-left side; moreover put the latin names of taxa in italics; precise also in  a caption below the figure the meaning of "only Monilinia..." and "only Prunus..." in (b), as apparently the figure concerns the two taxa (overlaps) and not only one;

10 in 3.4, indicate more clearly why you selected these Sichuan locations. Moreover, do you think that this selection is (significant) of the performance of the whole set of your data?

Reviewer 2 Report

The manuscript entitled Prediction of the potential distributions of Prunus salicina Lindl., Monilinia fructicola, and their overlap in China using MaxEnt, approaches an interesting, and quite current, topic, namely the application of statistical and mathematical tools for modelling ecological niches and their application to agriculture. 

However, the manuscript is still at a rather immature stage. The introduction does not precisely approach the state of the art, and does not support any starting hypotheses, and therefore the objectives of the work are not clearly and concisely stated.

The main problem is found in the material and methods. The application of a powerful tool such as modelling by means of the maximum entropy principle, implemented in the MaxEnt programme, has many advantages, and one main drawback, which is that it is extremely easy to use, but very difficult to know how it really works, so that the calibrations of the model imply a priori an expected result supported either by the users' knowledge or by some bibliographical source. 

Although the interface is relatively simple, the authors do not clearly explain how they have controlled various programme functions, such as the choice of logistic output (assuming certain predefined parameters which may not necessarily be appropriate for their study), nor do they clarify the type of validation or the number of replications used. It does not appear whether the model is over- or under-fitted. They do not take into account the regularisation values of the response curves of the variables, they do not clarify whether the model is optimised for a given background, neither does the number of background points used appear, and so on and so forth, functions that are beyond the authors' control. 

Although the use of the WorldClim database is becoming more widespread, in my own studies, the layers from this provider use an interpolation or spline-type surface generation methodology. When a geostatistical interpolation would be better. Therefore, the use of these databases is in many cases smoothed, and the authors should complete it with raw climate data, duly processed from the study area, to correct the WorldClim model. 

The authors also fail to explain in more detail the methodology used in the choice of the climate variables used.

Due to the authors' approach to the manuscript, I have not been able to continue reading the results, discussion and conclusions, because with the methodology used, there are many gaps that can reliably and realistically support the results obtained. 

I encourage the authors to thoroughly revise their manuscript, and to continue delving into this interesting topic, so that their work can be of interest to many people. 

Round 2

Reviewer 1 Report

Dear colleagues,  

I was very glad to read again this article, largely improved, however still minor points request a minor revision.

1 I still maintain my major point 3, especially for fungi (e.g. lines 49-51, and others…) in your revised parts. Please check latin names one by one throughout the text (for instance your “var nectarine” line 36 should be written “var. nectarine” as it is an abbreviation. Use international Plant Names Index (IPNI) https://www.ipni.org/) for plants, or equivalent. For fungi use https://www.mycobank.org/ or equivalent.

2 I still maintain too my minor point 9: in “new” figure 7, “precise also in a caption below the figure the meaning of "only” Monilinia... and "only” Prunus... in (b), as apparently the figure concerns the two taxa together and not “only” one ,it will be much more rapid to understand.

Yours very sincerely,

Reviewer 2 Report

First of all, congratulations to the authors for the great improvement in the manuscript. The manuscript is now much better understood, and the results seem robust to the methodology employed.

Based on the corrections made by the authors, we reviewers can now be more effective, and the results seem to be robust with the methodology used:

- It is advisable to use keywords different from those used in the title of the manuscript, this makes the search for potential readers more complete and agile.

- Check the formatting and editing of the text, such as the correct use of "[]" or "()".

- It is important that authors check the formatting of Latin names as well as the taxonomy names used. It is recommended to use the IPNI database. (e.g.: Prunus persica var nectarine (Ait) Maxim. the correct format is with the word "var." in non italics...). Another example: "Pyrus L.", the correct form would be to provide "Pyrus sp." with the genus name in italics, and it would not be necessary to provide the author's name, since the authorship of a taxonomic name is complete, including the specific epithet).

- At least, the first time the name of a taxon appears, it is necessary to provide the authorship.

In general, scientists must be neat in following the Code of Botanical Nomenclature, and be rigorous in that respect so that they know at all times which organisms they are working with.

- At the end of the introduction, they should write the starting hypothesis clearly and precisely (which is based on the background of the introduction). A somewhat exaggerated way would be: "Based on the background used, our starting hypothesis is ...". After providing the starting hypotheses, explicitly, come the different main and secondary objectives.

Writing in a clear, explicit and precise way helps to better understand the work, and to make it easier for a potential non-specialist reader to understand what the authors intend with their work.

- Lines 126-121: Revise the text format and heading of table 1.

- Line 134: What is IPCC, please define what it is the first time an abbreviation appears.

(For me, the IPCC is "The scientific group assembled by the United Nations to monitor and assess all global science related to climate change.)

- Lines 140-149: Check the formatting of the text.

- Lines 140-142: In that sentence, provide a reference. I disagree with that statement. AUC, only useful for comparing models with each other. Higher AUC does not imply better models in many cases. In overfitted models in MaxEnt, higher AUC values do not imply better models.

Results and discussion seem consistent and very interesting.

Conclusions: It would be good if the authors would specify more explicitly the conclusions of their work.

Congratulations on the manuscript.
